



# The effect of organic matter (OM) quality on the redox stability of OM-Fe association in freshwater sediments

Nana O.-A. Osafo*[1,2], Jiří Jan [1,3], Petr Porcal[1,2,3], Daniel A. Petrash[1,4], Jakub Borovec[1,2,3]

[1] Soil and Water Research Infrastructure, Biology Centre CAS, České Budějovice, Czech Republic

[2] Faculty of Science, University of South Bohemia, České Budějovice, 370 05. Czech Republic

[3] Institute of Hydrobiology, Biology Centre CAS, České Budějovice, 370 05, Czech Republic

[4] Department of Environmental Geochemistry and Biogeochemistry. Czech Geological Survey, Prague, 152 00 Czech Republic

*Correspondence to*: Nana O.-A. Osafo (nana.osafo@bc.cas.cz)

**Abstract.** Redox sensitive iron (Fe) oxyhydroxides participate in the biogeochemical cycling of nutrients and trace metals.
Their co-precipitation with organic matter (OM) leads to environmentally relevant OM-Fe associations. The stability of OM in such associations is still uncertain. It has been proposed that OM either increases the stability of the complex against reductive dissolution or enhance the dissolution (both abiotic and biotically) of Fe oxyhydroxides. The OM character, in terms of specific functional groups binding to Fe, could be a critical factor determining the fate and stability of OM-Fe. Here, sediment samples from a vertical profile of a mesotrophic freshwater reservoir were treated using a sequential extraction
scheme designed to distinguish Fe oxyhydroxides of different redox reactivity based on dissolution kinetics. To assess the reactivity and stability of the complexes, special attention was payed to the determination of redox active vs. stable fractions of Fe and the corresponding dissolved organic matter (DOM) components sequentially extracted. The quality of the extracted DOM was evaluated using UV-VIS and fluorescence spectroscopy together with the PARAFAC model. A selectivity model was used to ascertain the quality of OM preferentially associated with the different redox stable Fe oxyhydroxides.
Accordingly, we found that humic-like substances render the OM-Fe associations redox labile, while non-humic substances enhance the stability of the associations. These findings improve the understanding required for predicting the fate of OM-Fe associations in freshwater sediments with different organic matter sources and characters.

## 1 Introduction

Organic matter (OM) in lacustrine and soil ecosystems influences the coupled biogeochemical cycles of nitrogen (N) and iron

(Fe), as well as trace metal bioavailability. The association of OM, Fe and aluminum (Al) oxyhydroxides prevents the microbial degradation of OM and enhance its stabilization in both soils and sediments (Kaiser and Guggenberger, 2000; Keil et al., 1994; Lalonde et al., 2012; Mikutta et al., 2007; Nierop et al., 2002; Torn et al., 1997; Wagai and Mayer, 2007) From these associations, Fe oxyhydroxides are of paramount interest because their reductive dissolution is intrinsically linked to other biogeochemical cycles, including those of carbon and phosphorus.

The affinity of OM toward Fe and the stability of the resulting organometallic complex  is driven by several factors, with the most relevant being the structure and composition of the OM involved (Kellerman et al., 2015; Du et al., 2018; Han et al., 2019). Also important is the variable redox sensitive nature of Fe oxyhydroxides, with some of them being relatively more





soluble, and thus generally referred to as the active pool, while others, displaying a more stable soluble behavior are known as the less active pool. The latter could play a major role in stabilizing the OM and Fe association both in soil and sediments (van

der Zee et al., 2003; Keil and Mayer, 2014). OM being heterogeneous in composition and redox behavior (Neff and Asner, 2001) also adds to the complexity of efforts aimed at fully characterize and understand the nature of the OM-Fe association. Components of dissolved organic matter (DOM) that could increase its redox reactivity include quinone-hydroquinone moieties as these engage in the shuttling of electron with Fe oxyhydroxides (Aeschbacher et al., 2010; Garg et al., 2015; Klapper et al., 2002; Klüpfel et al., 2014; Scott et al., 1998; Struyk and Sposito, 2001).

The co-precipitation and, in general, the complexation of OM and Fe oxyhydroxides at transitional redox boundaries usually leads to the formation of inner sphere interactions (Keil and Mayer, 2014). Such interactions could result in the transformation of iron mineral phases to more crystalline structures, which may ultimately lead to conditions prone to the preservation of the organic matter (Eusterhues et al., 2008; van der Zee et al., 2003). Due to the heterogeneous structure of OM, there have been different conclusions in terms of which specific component determines or favor its co-precipitation/adsorption with Fe-

oxyhydroxides and their subsequent stabilization in soil and sediments (Lalonde et al., 2012; Adhikari et al., 2017; Adhikari and Yang, 2015; Du et al., 2018; Han et al., 2019). Identifying the organic matter component that more effectively stabilizes an OM-Fe complex is crucial for elucidating its fate. The OM-Fe complex has both been argued to form a rusty carbon sink in anoxic sediments (Lalonde et al., 2012) as well as responsible of inducing the cycling of elements such as N, P and As, which are interlinked to the dissimilatory reduction of Fe (Borch et al., 2005; Xiao et al., 2016).

To understand which organic compounds are selectively preserved by the OM-Fe association, a number of studies have used either synthetic OM or Fe-oxyhydroxides (Du et al., 2018). Few others have used natural sediments that were characterized by applying an extraction protocol that selectively dissolves the OM associated with Fe oxyhydroxides using dithionite as the extractant (Lalonde et al., 2012; Wagai and Mayer, 2007). However, essentially missing in all the previous analytical approaches is the definition of the variable reactive Fe pools (in terms of the redox lability) present in the sediment/soil

fractions evaluated.

Here we use an easily accessible and economic method to determine which OM components contribute to redox stability of Fe-OM associations in natural aquatic environments. We applied a sequential extraction scheme proved to dissolve the different natural Fe pools in terms of redox lability (active *vs.* more stable) Fe-oxyhydroxides (e.g., Jan et al., 2015) followed by spectroscopic characterization approaches of DOM that include UV-VIS and fluorescence techniques. In addition, we

compared bulk extractions to point out the advantage of sequential scheme in identification of the effect of OM quality.

## 2 Materials and Methods

### 2.1 Site description

Landstejn reservoir is a mesotrophic freshwater system located at the southern Bohemian region of the Czech Republic that serves as a potable water supply. The reservoir has a volume of $3.12 \times 10^6$ m$^3$, with a 25 m maximal depth and surface area of





0.4 km$^2$. Its annual average flow is estimated to be 0.09 m$^3$s$^{-1}$. The reservoir is surrounded mainly by forest and minor settlements.

## 2.2 Sampling

Four different sediment cores were obtained along longitudinal transects defined from the tributary to the dam section by using a gravity corer. Sediment cores were transported in the dark at 4 ˚C. Sediments were sliced in the laboratory into different
vertical layers, i.e., 0–0.5 cm, 0.5–1 cm, 1–2 cm, 4–5 cm and 10–12 cm. Sediments were homogenized and dry mass (105°C, 2 h) as well as loss on ignition (LOI, 550°C, 2 h) were determined. The proportion of dry mass increased along the vertical layers while the proportion of organic matter decreased marginally from the 2-12 cm layers, being more stable in the first 2 cm of the sediment pile.

## 2.3 Sediment extractions

First, each sediment sample was extracted sequentially following Jan et al. (2015) except for the HCl extraction step. Second, the samples were extracted using a bulk scheme (Fig. 1). The DOM extracted from water, the fractions from bicarbonate dithionite—a reducing agent, and the two fractions from NaOH are hereafter referred to as H$_2$O, BD1 and BD2, OH1 and OH2 respectively (Fig. 1). Fe is analyzed in the BD extracts. Differences in concentrations obtained in the sequential steps BD1 and BD2 and OH1 and OH2 are based on dissolution kinetics of both Fe and Al oxyhydroxides and their extraction efficiency (Jan
et al., 2015). Accordingly, the amorphous/poorly crystalline oxyhydroxides are dissolved in the shorter time steps (BD1, OH1), with the Fe extracted by BD1 being redox active, whilst the crystalline/stable oxyhydroxides were redox stable and dissolved in the longer time steps (BD2, OH2). In consequence, the Fe extracted by BD2 is referred to hereafter as less active. It must be noted, however, that a bulk sediment extraction with the longer time steps is not able to fractionate poorly crystalline from crystalline as it is the case in sequential extraction (Jan et al., 2015). Each of the mixtures were centrifuged at 3000 g for 10
minutes. The supernatant was then filtered using a 0.4-μm pore size glass fiber filter (MN GF-5, Macherey-Nagel, Düren, Germany). Extracts from (BD) were acidified to pH ranging from 3 to 5 units to keep the Fe in the dissolved form and then aerated for 2 h to reduce the influence of sulphur. H$_2$O, BD and NaOH extractants were used for extracting labile DOM, DOM-Fe and DOM-minerals, respectively (Lalonde et al., 2012; Lopez-Sangil and Rovira, 2013).

## 2.4 Analytical Methods

DOC was measured using a TOC-L CPH/CPN, Shimadzu with platina used as a catalyst and detected with an infrared detector. The concentration of Fe was measured in high energy helium gas, using 8800 Triple Quadropole ICP-MS, Agilent Technologies Accuracy and precision of the measurements were better than 3 standard deviations of analytical stock Fe solution. The molar ratio of DOC: Fe can then be calculated.





### 2.4.1 UV-VIS Spectroscopy

UV-VIS spectrum from 200 to 800 nm with step of 1 nm was measured in 5 cm quartz cuvette (UV – 2700, SHIMAZDU, Japan). The absorption coefficient at 254 nm (mathematically given as $SUVA_{254}$ = (Sample Absorbance – Blank Absorbance)/ path length (m)]) was normalized with the respective DOC (Absorption coefficient/DOC (mg $L^{-1}$) for $SUVA_{254}$ as a proxy to aromaticity (Weishaar et al., 2003). The slope ratio (SR), which is a proxy to the molecular weight, was estimated as the ratio of the slope of absorbance curve at shorter wavelength (275–295 nm) and longer wavelength (350–400 nm)(Helms et al.,

2008). Though the SR was originally water fractions, the ratio has been also applied to evaluate other fractions (He et al., 2016).

The presence of Fe in sample can influence the absorption coefficient at 254 nm (Poulin et al., 2014). Such effect of Fe on $SUVA_{254}$ was eliminated by subtracting it with the aid of a linear calibration curve between $A_{254}$ and Fe concentrations (Fig. S1). The DOC for the Fe concentrations was in the range of $0.60 \pm 0.02$ mg $L^{-1}$.

### 2.4.2 Fluorescence spectroscopy

Excitation and Emission Matrices (EEM) of fluorescence DOM (FDOM) were measured using a FluoroMax3 (Horiba, Japan) instrument with excitation wavelengths in the range 240-500 nm with a 10 nm increment step, and emission was 270-620 nm with an increment of 2 nm. To compare the fluorescence intensities among fractions, the specific fluorescence intensity (SFI) was estimated as measured fluorescence intensity (R.U.) divided by DOC concentration (mg $g^{-1}$). The concentration of Fe did

not affect the SFI results due to low pH of BD extracts as was already shown before (Poulin et al., 2014) .

### 2.5 Data Analysis

Four sediment cores were sampled from the reservoir. Out from these four cores the mean values and the standard deviation of each parameters were estimated to construct a vertical profile. A Student t-test was used to estimate the significance ($p <$ 0.05) of the difference between BD1 and BD2.

A parallel factor analysis (PARAFAC) was done in R studio (version 3.5.3) using staRdom package (Pucher et al., 2019). A 3 components model was obtained for all extractions and interpreted using the online spectral library of organic compounds in the environment OpenFluor (Murphy et al., 2014; https://openfluor.lablicate.com/), using a spectral match better than 95%. To determine the components that preferentially associate with the different pools of Fe oxyhydroxides extracted, we used the following Eq. (1) for selectivity, Sel (in percentage) calculated following:

$$\text{Sel } (\%) = 100 * \left( \frac{C_F}{C_{AS}} - 1 \right) , \tag{1}$$

where where the ratio $C_F/C_{AS}$ is the component content in each fraction divided by the mean content of that component in all extracts. The mean was estimated from all the extraction steps for the respective component along the vertical profile. Components selectivity was thus ascertained with reference to the mean component from all extraction steps. The expression



for the mean equals zero. A positive value indicates that a given extractant extracted the component more efficiently compared
to the mean extractants, hence there is component selectively associated to that extractant. Conversely, a negative value of this
expression indicates that the extractant extracted the respective component at a lower intensity with reference to the mean,
thus the extract exhibits no selectively toward any given component.

## 3 Results and Discussion

### 3.1 DOC and Fe concentrations

The concentration of dissolved organic carbon (DOC) extracted by individual extractants in the sequential extraction scheme
ranged between 0.8–39.3 mg g$^{-1}$ (Fig. 2). DOC concentrations from H$_2$O extracts were the least significant, consistently
yielding low contents of dissolved or loosely bound organic carbon (0.9 ± 0.4 mg g$^{-1}$). Conversely, OH-extracted extracted
most DOC, 36.3 ± 7.4 mg g$^{-1}$ and 22.1 ± 7.2 mg g$^{-1}$ in OH1 and OH2, respectively. About 13 % from the total extracted, average
DOC was estimated in BD extracts, 5.0 ± 1.9 mg g$^{-1}$ and 3.8 ± 1.1 mg g$^{-1}$ in BD1 and BD2, respectively. This fraction represents
DOM associated to Fe oxyhydroxides (Lalonde et al., 2012; Wagai and Mayer, 2007). The corresponding Fe concentrations
were 8.0 ± 3.6 mg g$^{-1}$ and 2.9 ± 1.5 mg g$^{-1}$, which results in DOC:Fe ratios of 3.0 ± 0.4 and 6.9 ± 1.8 for BD1 and BD2,
respectively. (Fig. 3b-c). DOC concentrations in BD1 were statistically higher than DOC in BD2 (p < 0.002), which implies a
higher affinity of OM for the active pool of Fe mineral oxyhydroxides (Fig. 3a).

Figure 3a shows that along the vertical sediment profile, the proportion of DOM bound to the active pool was significantly
higher than that bound to the redox stable pool. It was observed, however, that the DOC associated with these stable phases
remained unchanged along the vertical profile, whilst the DOC fraction linked to the redox active phases declined. This result
implies a high reactivity, in terms of sorption and desorption, toward the sediment water interface as compared to deeper in
the sediment pile. This could be linked to utilization/mineralization of fresh biomass initially associated with the active or
poorly crystalline Fe-phases.

The co-precipitation/adsorption of DOM and Fe can be estimated by using the DOC:Fe molar ratio. However, the interpretation
of this parameter regarding the degree of preservation remains ambiguous. A school of thought argues that the lower the
magnitude the higher the preservation (Chen et al., 2014, 2016), whilst another argues otherwise (Han et al., 2019). Ranging
between 1.2 and 10, the DOC:Fe ratios in our sediment profile (Fig. 3c) were consistent with previously reported ranges (Chen
et al., 2014; Lalonde et al., 2012). The DOC:Fe for BD1 were significantly lower than BD2 (p < 0.001). DOC:Fe for BD1
were in the range 2.9-3.3 whilst BD2 ranged between 6.3-7.4. From this results, it is inferred that higher DOC:Fe values are
associated with more stable, crystalline Fe minerals, with that pool being representative of the stabilized/preserved DOM. This
is in agreement with the recent work by Han et al. (2019), which reports that comparatively higher DOC:Fe values correspond
to stable OM-Fe complexes. Nonetheless, in our study site we also observed that important variations in the sediment' DOC:Fe
ratio along the longitudinal profile. These can be linked to a substantial increase (4-fold) of Fe concentrations from the tributary





to the dam. Alternatively, the longitudinal shift could be related to changes in the reactivity DOM toward excess Fe as the former change molecular size and composition during transit through the reservoir.

## 3.2 UV-VIS Spectral Characteristic of Extractants

### 3.2.1 SUVA$_{254}$

Specific ultraviolet absorbance of DOC at the 254 nm wavelength, SUVA$_{254}$, is an index used as a proxy for aromaticity that
for our individual fractions ranged from 1.6 to 5.3 L (mg C) $^{-1}$ m$^{-1}$ (Fig. 4a). Higher values correspond to the OH1 fraction-extracted DOM (5.0 ± 2.2) whilst the OH2-extracted DOM has SUVA$_{254}$ values of 1.8 ± 0.4. With the Fe effect subtracted, both H$_2$O' and BD' SUVA$_{254}$ values were comparable, with no significant difference in the SUVA$_{254}$ values of DOM extracted by BD1 and BD2. In consequence, these two fractions are thought similar in their aromatic yield.

### 3.2.2 Slope Ratio (SR)

For individual fractions, the SR, proxy for DOM molecular weight ranged from 0.9-5.1. From the literature, SR more typically ranges from 0.4-12.0 (Helms et al., 2008; He et al., 2016). The higher the SR values the lower the molecular weight (Helms et al. 2008). Higher SR values in sequential extraction correspond to DOM extracted by BD2 (i.e., 6.0 ± 1.7), followed by BD1 (2.2 ± 1.4). The SR of DOM extracted by H$_2$O, OH1 and OH2 were 1.6 ± 0.4, 1.4 ± 0.2 and 2.8 ± 0.7, respectively (Fig. 4b). Though there was no substantial difference in aromaticity of the DOM extracted by the BD1 and BD2 fractions, their SR
differed significantly. The SR value of DOM from BD2 in sequential extraction was significantly higher compared to that from BD1 (p = 0.01), meaning that BD2 extracted comparatively lower weight aromatics. This implies that the aromatic structure of the DOM extracted by the two BD steps may differ despite their similar SUVA$_{254}$ score range. Such structural difference could be ascribed to the number of aromatic rings. This result is important as it indicates that DOM associated with different redox Fe oxyhydroxide' pools vary in terms of quality and lability.

## 175   3.3 EEM-PARAFAC

Three components allowed us further elaborating on the chemical structure of the FDOM as determined from their maxima excitation and emission wavelengths in PARAFAC and interpreted by using their loadings in the openfluor model (Fig. S5). The range of intensities of these three components, namely C1, C2 and C3, are 0.1-6.0, 0.1-1.4 and 0.0-3.5 RU g mg$^{-1}$ C, respectively (Fig. 5). The H$_2$O fraction extracted DOM with the highest SFI in all components, which ranged from 5.9 ± 1.3,
1.1 ± 0.3 and 3.3 ± 1.0 RU g mg$^{-1}$ C for C1, C2 and C3, respectively. The intensities associated to the OH-extracted FDOM fractions were for all components ≤ 0.2 mg g$^{-1}$. The SFI of FDOM extracted by BD1 was 0.8 ± 0.3 RU g mg$^{-1}$ C for C1, and 0.3 ± 0.1 RU g mg$^{-1}$ C for both C2 and C3. Finally, the FDOM extracted by BD2 was 1.0 ± 0.4, 0.3 ± 0.1 and 0.7 ± 0.4 RU g mg$^{-1}$ C for C1, 2 and C3 respectively.



Table 1 listed results of openfluor database for the 3 model components. Accordingly, with an excitation to emission ratio

(Ex:Em) of 240/454, Component 1 (C1) could be interpreted to be the humic-like substance fulvic acid. C1, however, also falls in the range attributed to reduced quinones (Ishii and Boyer, 2012). The longer wavelength in the emission range of C1 has been correlated to a high molecular weight and hydrophobic nature (Wu et al., 2003). The component can be sourced from both microbial and terrigenous sources (Lapierre and Frenette, 2009; Stedmon and Markager, 2005).

Component 2 (C2) has an Ex:Em of 310/394, which is generally reported as a humic-like substance exhibiting less aromatic

rings and hydrophobicity and lower molecular weight than C1 (Wu et al., 2003). From its Ex:Em, this component is also known to possess characteristics of oxidized quinones and can be sourced either from both terrestrial and microbial organic matter (Ishii and Boyer, 2012). Finally, component 3 (C3) exhibits an Ex:Em ratio of 270/316 that can be ascribed to a protein-like moiety. In terms of the OH-extracted FDOM fraction, their near zero SFI values for C3 indicate that DOM associated with minerals are protected from microbial degradation ( Keil et al., 1994; Lalonde et al., 2012; Rothman and Forney, 2008).

For both BD fractions, larger variations in the vertical profile of FDOM were seen in C3. Iron oxyhydroxides are said to have higher and moderate affinity to C1 and C2 (Ishii and Boyer, 2012). It is argued that the association of DOM to oxyhydroxide phases could reduce biodegradation of the former through the formation of stronger bonds, distortion of the crystal structure of the oxyhydroxides and formation of larger molecules (Eusterhues et al., 2008; Mikutta et al., 2007). This seems to be reflected by our results as low SFI values of C3 from the OH fraction generally represent FDOM bounded to other mineral

phases, such as clays or perhaps Al-hydroxides, while the DOM-Fe association extracted by the BD sequential steps had comparatively much higher C3' SFI values (Fig. 5).

The previous result and its implications to DOM preservation could also be speculatively linked to an electron shuttling intrinsic to the Fe-DOM complex. Chemolithotrophic microbes capable of conducting OM respiration process coupled to iron utilization could be using C1 and C2 as an electron donor or acceptor, especially because these components may also

correspond to reduced and oxidized quinone moieties, respectively (Li et al., 2014; Orsetti et al., 2013). Under conditions of transient anoxia usually observed in aqueous settings, these components can be regenerated and recycled (Aeschbacher et al., 2011, 2012). We further elaborate on this alternative avenue for DOM preservation below.

### 3.4 DOM component preferentially associated with Fe

The long-term preservation of DOM-Fe in sediments is generally thought to be enhanced by the irreversible reactivity of the

DOM component and Fe-oxyhydroxides (Henrichs, 1995). The structure of both C1 and C2, however, possesses a redox behavior that makes their reaction with Fe-oxyhydroxide reversible leading to the solubilization of the DOM or both Fe and DOM, with the latter subsequently being biodegradable (Burdige, 2007; Keil and Mayer, 2014). Yet, C3 does not contain a redox component, and, thus, its reaction with Fe hydroxide minerals is irreversible.

A selectivity model mathematically described in Equation 1 (Section 2.5 above) was used to study the components that are

preferentially extracted by all the extracts, with emphasis on the sequential BD 1 and BD2 steps (Fig. 6b). It was observed





that C1 and C2 were selectively extracted by BD1, which could also explain the different SR values of the BD extractions. Wu et al. (2003) observed that OM components emitting in longer wavelengths have comparatively higher molecular weight compared to those emitting in shorter wavelength. C2 in the first 2 cm of the sediment pile, however, was selectively dissolved by BD2. Conversely, C3 was only selectively extracted by BD2 and it increases with depth along the vertical profile.

From our modelling results, it could be inferred that the regenerable humic-like components (C1 and C2) have high sorption affinity to Fe mineral oxyhydroxides i.e. both C1 and C2 react to a large extent, faster with Fe (III)-bearing minerals (e.g., Du et al., 2018) especially in the redox active pool that include the poorly crystalline oxyhydroxide phases. However, the stability of the DOM-Fe, which enhances the long-term preservation of DOM-Fe, is also influenced by C3 as observed in other studies (Adhikari and Yang, 2015; Lalonde et al., 2012) and this we found to rather occur in the redox stable/less active pool (BD2)

of the reactive Fe-oxyhydroxide mineral phases. This observation agrees with the assertion that irreversible reactions enhance the stabilization of the DOM-Fe at longer timescales. In this regard, our interpretation is at odds with Han et al. (2019) whose study also concluded that high DOC:Fe values could be an indication of stability in the DOM-Fe but concluded the OM stabilised in the association was mainly humic-like in character. The results dissimilarity could result from synthetic Fe-oxyhydroxides used in the latter study are not exposed to environmental conditions over an appreciable time. Further work is

thus required to conclude on this as the quality of the DOM as well as its loading (DOC:Fe) might influence the dissolution kinetics (Eusterhues et al., 2014).

## 4. Conclusions

Humic-like substances of DOM were selectively associated with the redox active Fe pools whilst the stable Fe pool selectively associated with proteinaceous (non-humic) substances. As the humic-like substance is known to contain quinone moieties that

makes them redox active, their binding with iron undergoes a reversible reaction whilst that is not the case for proteinaceous substances. We therefore determined that the rusty carbon sink by OM-Fe is mostly comprised of the non-humic substances of DOM and a redox stable Fe oxyhydroxides. Accordingly, this study is significant as it addresses the gap due to studying bulk Fe oxyhydroxides in DOM-Fe without distinguishing redox stability. Importantly, as indicated our results are consistent with other studies about the quality of OM stabilized in OM-Fe association, yet we achieved them by using rather simple

methods. In addition, the sequential extraction used in our study gives an indication of the time scale of stability of OM-Fe when bound by humic as compared to non-humic components. Finally, by applying our characterization approach, it is possible to estimate the fate of OM and its role driving environmentally important cycles connected to Fe(III) as this is used as an terminal electron acceptor and thus shaping interactions between the biological, chemical, and physical processes in aqueous settings. Due to many aspects linked to global warming, upcoming changes in the source, quality, and concentration of

dissolved organic matter fluxes to the aquatic environment make our observations relevant for environmental characterization. Below we summarized our observations as concluding remarks:



- Sequential BD1 and BD2, is vital for estimating DOM-Fe in terms of how stable or redox active Fe oxyhydroxide is which cannot be said in bulk BD extraction.

- A higher DOC:Fe molar ratio corresponds to the complex stability as shown by differences between BD1 and BD2.

- No substantial difference in SUVA$_{254}$ for BD1 and BD2 indicates similar DOC character in terms of aromaticity for both fractions. However, significantly higher SR for BD2 indicates that DOC with lower molecular weight in complex with Fe increase redox stability.

- PARAFAC model of the FDOM in the different fractions indicate high reactivity of Fe toward humic components, yet the stability of the complex is given by non-humic component (C3).

- Humic-like substances of DOM might have high reactivity towards Fe oxyhydroxides, however the complex formed is stabilized by the non-humic substances.

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

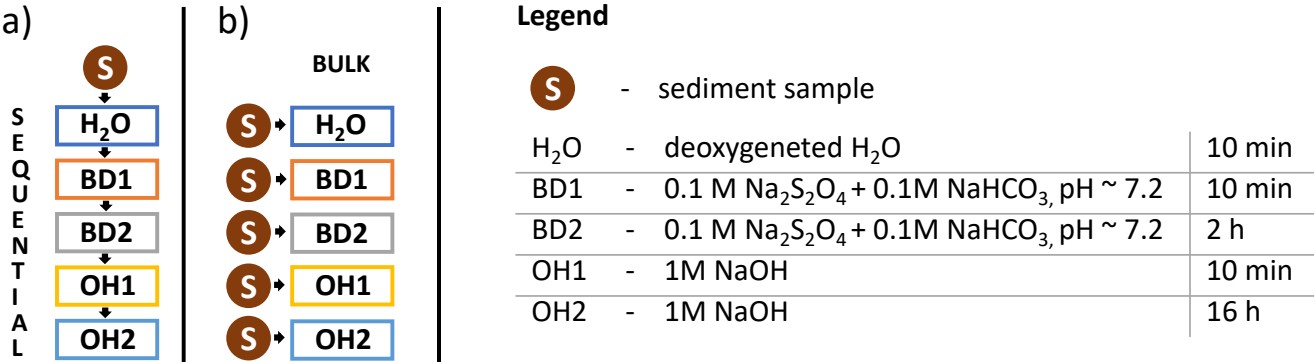

Fig. 1. Scheme for sequential (a) and bulk (b) extractions. The legend describes the composition of extractants and extraction times.






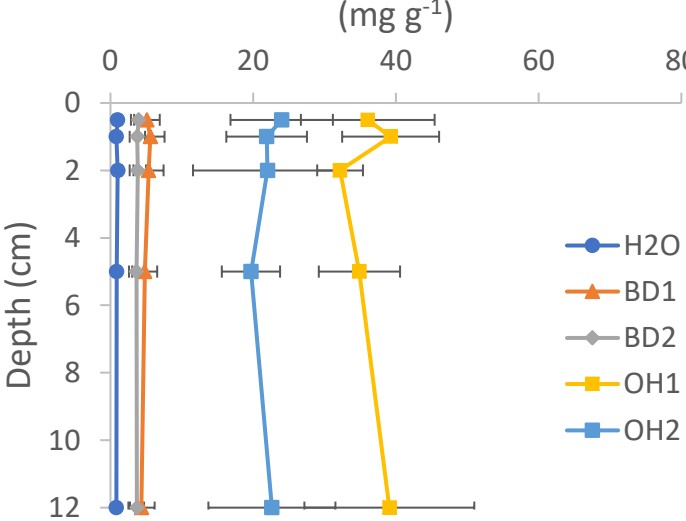

**Fig. 2. Mean concentrations of extracted DOC from sequential extraction. Values are relevant to means and SD of four sediment cores along the vertical profile of Landstejn reservoir.**



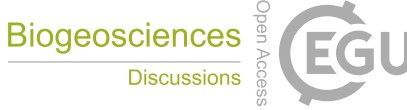

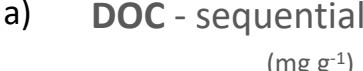

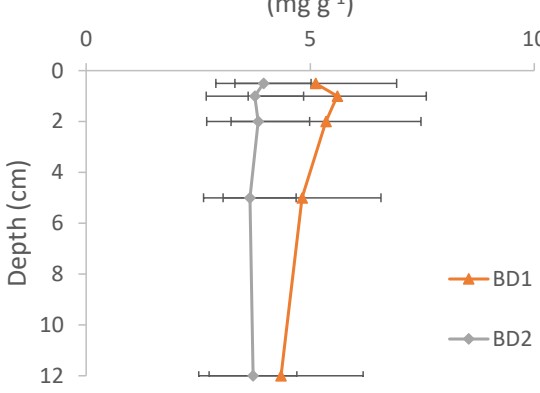

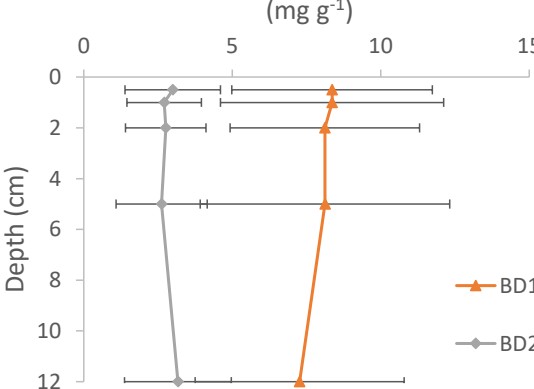

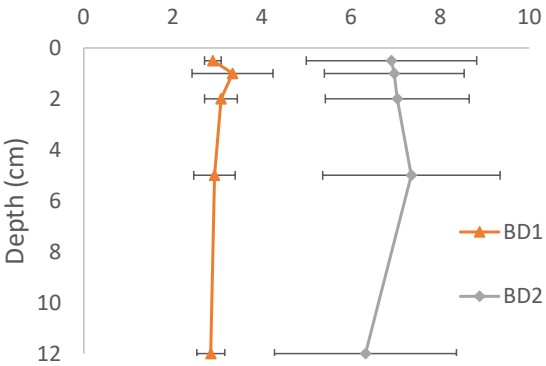

**Fig. 3. Concentrations of DOC extracted by BD1 and BD2 in sequential (a), and Fe extracted by BD1 and BD2 (b) and molar ratios of DOC and Fe extracted by BD1 and BD2 in a sequential scheme (c). Values are relevant to means and SD of four sediment cores along the vertical profile of Landstejn reservoir.**





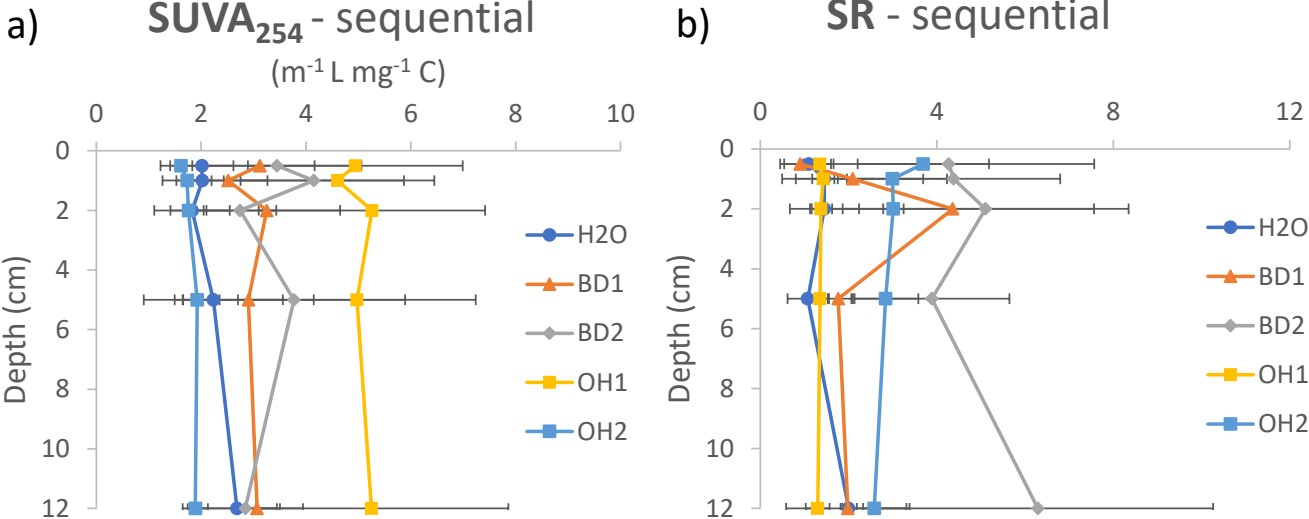

**Fig. 4. Values of SUVA$_{254}$ used as a proxy to aromaticity (a) and SR proxy to molecular weight (b) for individual sequential extractions. Values are relevant to means and SD of four sediment cores along the vertical profile of Landstejn reservoir.**



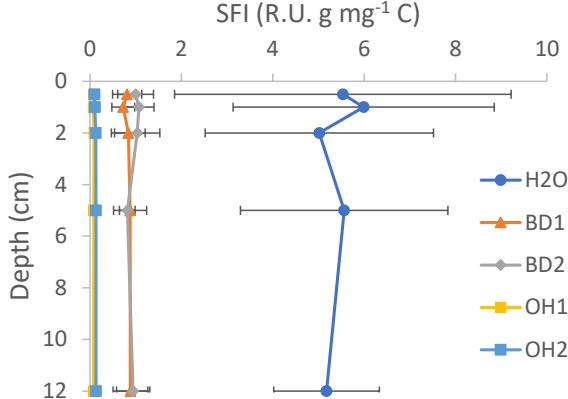

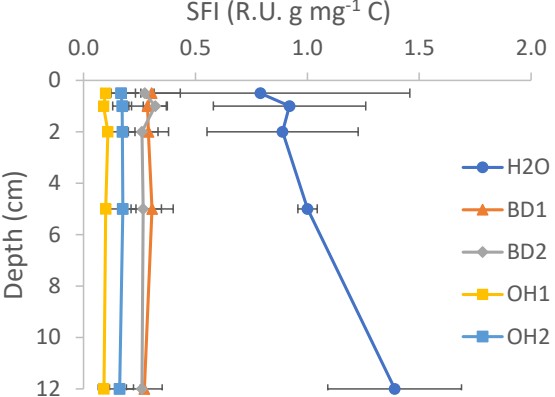

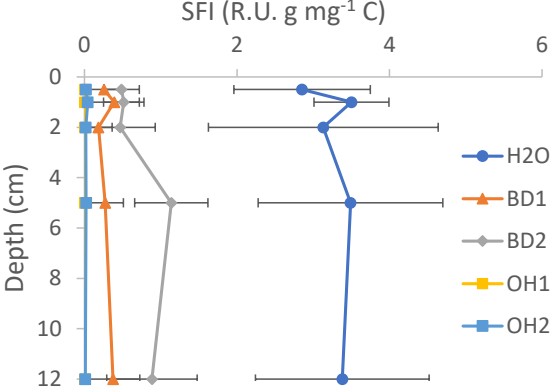

**Fig. 5. Specific fluorescence intensities– SFI (given as the ratio of the fluorescence intensity in R. U. to the DOC concentration in mg g$^{-1}$) of identified components (C1, C2 and C3) for individual extracts in sequential extraction scheme. Values are relevant to means and SD of four sediment cores along the vertical profile of Landstejn reservoir.**



Fig. 6. Mean proportions of the different PARAFAC components (C1, C2, C3) from all extracts in sequential extractions along the vertical profile of four sediment cores (a) and a selectivity model for sequential BD1 and BD2 (b). The selectivity model shows relative changes against the mean of all extractants with positive values (+) indicating preferential association of particular components in BD1 or BD2.





**Table 1. Summary of PARAFAC model component from openfluor database, similarity score of identified components with respect to cited studies and their description.**

| Comp. | Similarity score | References | Description |
|---|---|---|---|
| **C1** Max Ex/Em 240/454 | 0.99 0.99 0.99 0.99 | Osburn et al., (2016) Shutova et al., (2014) Bittar et al., (2015) Gonçalves-Araujo et al., (2016) | - fulvic acids - presence and origin mainly in terrigenous environment - also observed to be autochthonously sourced |
| **C2** Max Ex/Em 310/394 | 0.99 0.99 0.99 0.99 | Hambly et al., (2015) Kothawala et al., (2014) Schittich et al., (2018) Kowalczuk et al., (2009) | - humic materials processed by microbial activity - sourced both in terrigenous and aquatic systems |
| **C3** Max Ex/Em 270/31 | 0.97 0.96 0.98 | Bittar et al., (2015) Catalá et al., (2015) Yamashita et al., (2010) | - protein like components - mainly autochthonously sourced |

425