# Peer review of "The effect of organic matter (OM) quality on the redox stability of OM-Fe association in freshwater sediments"

_Biogeosciences, 2020_

## Referee Comment (RC1) · Anonymous Referee #1 · 24 Sep 2020

In this manuscript the authors aim to understand how OM-Fe associations respond to different chemical treatments in lake sediments, in order to improve our understanding of the fate of OM and Fe in freshwater sediments with different sources and characters. To do so, they characterized OM concentration and composition (using optical approaches), as well as Fe concentrations, for 4 sediment cores taken in a reservoir found in the Czech Republic. They compared the effects of different chemical treatments (water, acid, base) over different times, and found that their simple method allowed to extract different amounts and characters of OM and Fe for sequential and bulk samples. They found that large differences among treatments, and conclude that humic-like DOM was associated to active Fe pools whereas protein-like DOM was associated stable Fe pools. They argue that these findings have implications for global change given the expected changes in source, concentrations and quality of DOM fluxes to aquatic environments.

I found the topic of importance, but that the study was insufficiently developed to convincingly support the main conclusions of the manuscript. The manuscript is very short and does not deeply develops the core ideas in the introduction and discussion, and is hard to read in several instances. Overall I found that there was a mismatch between the broad conclusions on the role of the rusty OM sink in the face of global change vs the very specific information provided from 4 cores of a single reservoir (which are some times interpreted as replicates, some times interpreted as 4 independent samples on a spatial gradient). Moreover, given the lack of environmental gradients covered in these 4 cores, it is hard for the reader to appreciate how the authors results convincingly contradict previous findings on the role of DOM composition on the Fe-OM associations. Below I provide more detailed comments.

1- The introduction is very short and very specific, and it is unclear what the broadly interesting knowledge gaps are, and whether the study addresses fundamental knowledge gaps or hope to confirm known patterns with a simpler approach

2- Section 2.3: More details are needed to appreciate the chemical treatments, and what they mean for adequate interpretation of the results. I acknowledge that a reference is provided, but at least the reasoning behind the treatments is needed in this manuscript to understand the coming results. This is important because there is barely any difference among cores (or this is not shown), and from surface to bottom of the cores; most of the variation comes from across treatments

3- L100: I did not understand this sentence, and what the reference actually refers to

4- L102-104: I did not understand what was done

5- L109: But does not pH affect fluorescence?

6- Paragraph at L131: There is a huge difference among treatments, but without further explanation of what they mean, in the Methods, it is hard to appreciate these findings

7- L154-156: Where is this coming from? No result, or description of the cores was provided to appreciate this statement

8- Section 3.2.1: This section is too superficial and is barely interpreted

9- 3.2.2: same

10- 3.3: Here the authors go at length in describing the PARAFAC components, with a level of details that greatly differs from previous sections. I believe the idea is to interpret how specific components may mean a particular chemical composition, which is then used in the following section to interpret associations with Fe, but the ideas are not explicitly connected in my opinion; there is a leap in the level of information provided by the optical approaches in section 3.3 vs the detailed chemical interpretation provided in section 3.4. For example, on what basis, specifically, can you conclude the following: "The structure of both C1 and C2, however, possesses a redox behavior that makes their reaction with Fe-oxyhydroxide reversible leading to the solubilization of the DOM or both Fe and DOM, with the latter subsequently being biodegradable (Burdige, 2007; Keil and Mayer, 2014).". I am not saying that this interpretation is wrong, but I believe that there is not enough information in the manuscript that supports this interpretation based on the authors findings.

11- L223-225: I do not think these studies have identified C3 specifically (at least they are not presented in Table 1), so this wording is misleading

12- L228-229: What is "appreciable time", and why is it important? Also, it is weird to end the discussion saying that in the end we are not sure what is going on hence more studies are needed, I would suggest concluding on a more "positive" note in terms of your main contributions to this topic.

13- The conclusion overall is interesting to read, but I found it unsupported by the

results and insufficiently developed in key places of the manuscript. I suggest using sentences from the conclusion as key points (maybe paragraphs topics) that guide the development of the introduction and discussion

———————————————

---

## Short Comment (SC1) · 23 Oct 2020

We thank the reviewer for his/her valuable suggestions for improving our research report. At this point we would like to clarify to the audience of BG Discussions a few aspects:

The preprint under consideration evaluates previous work on both synthetic and natural samples in terms of the association of organic matter (OM) and iron (Fe) oxyhydroxides (OM-Fe). From those studies, there were disparity in concluding remarks on OM quality selectively preserved by Fe. In consequence, we implemented a simplified extraction scheme and analytical protocol that allowed us assessing OM quality associated to

the corresponding metallic oxide, and discriminate Fe in terms of its redox lability and stability as we believed the redox stability of Fe plays a role in the preservation. Our approach is intended to gain further insight on the role in the selective preservation of OM-Fe and bridges the gap existing in the current literature.

An aspect that seems to be a misunderstanding, however, is that our dataset is not discussed to the light of spatial gradient, or other specificities of the site where the cores were retrieved. Slight variations in [Fe] could arise from the site' configuration and hydrochemical gradient, and as such, it was mentioned. We will punctually expand the text to better explain the reasoning behind our deductive approach on how it is not affected by the specific site hydrochemical gradient.

In terms of the conclusions and their relevance to results presented, the main results refers to non-humic quality of OM being preferentially preserved on a longer time scale compared to humic components and this can be seen when a sequential extraction scheme is used instead of a bulk extraction, which does not necessarily differentiate redox labile Fe from redox stable Fe in natural environmental samples. We will highlight this results and its implications.

Other aspects that are unclear, as kindly noticed by the reviewer, are the pH at which the extraction of the reactive oxide phases (i.e., circumneutral; see Table 1), or the relevance of speculative statements regarding global environmental change and the role of OM-rusty sink in connection with GHG. The text will be edited for clarity and statements rather speculative and not sustained by the data will be removed in the next iteration of the report.

Again, we thank the reviewer for kindly highlighting flaws in the report, her/his the relevant suggestions for improving the discussions, and for pointing out the instances where the text needs to be edited for the sake of clarity or to prevent speculation.

Sincerely,

[Figure]

Nana O-A. Osafo

---

## Referee Comment (RC2) · Anonymous Referee #2 · 29 Oct 2020

This manuscript examines organic matter-Fe interactions in freshwater sediments with the goal of better understanding the fate of these organic matter-Fe associations. Interest here stems, in part, from recent work examining the "rusty iron sink" for organic matter preservation in sediments. This is an important area of research and I think there may be some interesting data here. However, the manuscript is too speculative in too many places, and a number of aspects of the work need additional clarification. For these reasons, this manuscript will need extensive revisions before it be reconsidered for publication.

One of the big problems I have with this manuscript is that it discusses processes

associated with iron redox cycling, but there is no evidence of such processes in the profiles presented here. For starters, virtually all depth profiles show no change with depth (especially given the associated error bars), and so I see no evidence, for example, of DOM or Fe solubilization in association with the processes proposed here. There is also no background information about the sediments to help interpret these profiles. For example: Are the bottom waters of the reservoir oxygenated, seasonally anoxic or permanently anoxic?; If there is oxygen in the bottom waters what is the oxygen penetration depth in the sediments, and does it vary seasonally?; What are the concentrations and depth profiles of dissolved iron in the sediment pore waters? With this kind of information one could begin to estimate where the sediment redox boundary is, which is critical for understanding the basic aspects of iron redox cycling in these sediments, and which could then aid in the interpretation of these results.

Listed below (by line number[s]) are some more specific concerns I have.

1. (71-3) – These are results and don't belong in this section. Also what does "more stable" mean?

2. (76) – Why is the bulk scheme mentioned, since as far as I can tell all of the data discussed here comes from the sequential extraction procedure.

3. (78- ) – I may be missing something here, but this sequential extraction scheme does not make sense to me. As I read the text, BD1 extracts Fe that is redox active, then BD2 next extracts iron that is redox stable, and then OH1 next extracts iron that is redox active and finally OH2 extracts iron that is redox stable. If these are sequential extractions and BD1 does not remove all (or most) of the redox active iron why, for example, does any redox active iron escape extraction during BD2 (which is the same as BD1 just longer) to then be extracted by OH1? What am I missing here?

4. I'm trying to compare this extraction scheme to others I am a bit more familiar with (e.g., Goldberg, et al. 2012. Chem. Geol. 296, 73-82; Poulton and Canfield 2005. Chem. Geol. 214, 209-221) and am having trouble understanding how NaOH extracts

iron oxides. Related to this, it's also not clear to me if any iron is actually extracted in the OH extractions – no data is presented.

5. The concept of redox "active" versus redox "stable" is a bit misleading. All iron oxides will undergo reductive dissolution, but the rates will vary by quite a bit (see, for example, Table 4.1 in Raiswell and Canfield (2012, Geochemical Perspectives 1, 1-220)).

6. The vast majority of the comparisons made later on in the paper are between the BD1 and BD2 phases (see, for example, the concluding remarks starting on line 247), thus I am further confused by where (and how) the OH extractions fit in here.

7. (137) – Differences in DOC concentration in the BD1 versus BD2 extracts do not necessarily imply differences in OM affinity for these different iron pools. For starters, there is more iron in the BD1 pool so there are presumably more potential iron oxide binding sites for DOM. That may be the simplest explanation for these DOC differences.

8. (154) – By averaging the results from the 4 cores you are implicitly assuming there is no spatial variability in the sediments along this sampling transect. However, here the authors talk about variations along this longitudinal profile (although no data are shown to support this assertion). Nonetheless, you can't have it both ways. If there is longitudinal variability among the cores you shouldn't be averaging the depth profiles and in fact, by doing so you may be obscuring real depth trends in each core.

9. (210 – 215) - This is far too speculative and not well supported by the data. Components C1 and C2 may include quinones that can react with iron oxyhydroxides and this may release DOM that may be biodegradable. Yet the profiles presented here show no evidence of this. Likewise, C3 may be non-redox active and may be irreversibly bound to the iron phases, but I see little to really support this assertion. The points made here in the text are also made in the Abstract (lines 20-21) and the Conclusions (lines 233-6), and are, in my opinion, presented in far too definitive a fashion, given the data presented here. Furthermore, generalization of this speculation to the "rusty iron sink" (line 236) is very premature (at best).

10. The differences between the fluorescence characteristics of the DOM associated with the different iron phases is intriguing, but in the context of all of the other issues I have with this manuscript, it's hard for me to be know how to interpret their significance.

---

## Author Comment (AC1) · 18 Nov 2020

General response

We appreciate the thorough work by the reviewer, and we share in the concerns raised. We hope our response would help put the study in perspective and thus bring clarity to the study and the reviewer. The study considered works which has been done so far with both synthetic and natural samples in terms of the association of Organic matter (OM) and Iron (Fe) oxyhydroxides (OM-Fe). From those studies, there were disparity in concluding which of the OM quality is selectively preserved by Fe. The study used a sequential extraction scheme which mainly shows form of OM quality

associated to a corresponding Fe oxyhydroxide and discriminate Fe in terms of its redox lability and "stability", since we thought that could play a role in the selective preservation of OM-Fe and is missing in literature reviewed so far. Thus, the quality of OM associated with the redox "stable" Fe are therefore the selectively preserved on a longer time scale. We would like to point out that the data for this study was not discussed in the light of spatial gradient. The only time that was used, was to explain the variations in Fe concentrations in the study. The study was mainly more on qualitative and mechanistic means of OM-Fe associations from 4 cores. In terms of the conclusion, the main conclusion was to show that non-humic quality of OM are preserved on a longer time scale compared to humic components and this can be seen when a sequential extraction scheme is used instead of a bulk extraction, which doesn't necessarily differentiate redox labile Fe from redox stable Fe in natural environmental samples. Also, we acknowledge why the reviewer interpretated the extraction as water, acid, base. However, we would like to state clearly that none of the extraction was done in an acidic medium. The dithionite was buffered to a circumneutral pH as seen in Table. 1 and the supernatant was only acidified after extraction to keep the Fe in dissolved form. With reference to OM-rusty sink in connection with changing of the climate, that was meant as an implication of this knowledge, for instance the thawing of permafrost soils that become water-saturated and reducing. Also, the main OM sources of a system could be used to predict how susceptible they could be in the mineralization of OM in producing greenhouse gases.

Specific response R1Q1 The introduction is very short and very specific, and it is unclear what the broadly interesting knowledge gaps are, and whether the study addresses fundamental knowledge gaps or hope to confirm known patterns with a simpler approach

A1. We thank the reviewer for pointing out concerns with the introduction. We take seriously the concerns of the reviewer. For clarity sake, the introduction sought to show the work done and the different conclusions as in the quality of OM been selectively

preserved. The reasons we assigned to this was due to the heterogenous nature of OM and hence the approach used in studying could be what is leading to the disparities in conclusion. Also, we concluded that none of the studies so far considered the different redox lability of Fe in their study. To that effect our work sought to use a sequential extraction scheme which discriminate in terms of redox labile and redox stable Fe which in our opinion is lacking in the studies so far.

R1Q2 Section 2.3: More details are needed to appreciate the chemical treatments, and what they mean for adequate interpretation of the results. I acknowledge that a reference is provided, but at least the reasoning behind the treatments is needed in this manuscript to understand the coming results. This is important because there is barely any difference among cores (or this is not shown), and from surface to bottom of the cores; most of the variation comes from across treatments.

A2. The main purpose of this extraction was to estimate OM associated with reactive Fe oxyhydroxides of different redox lability which is achieved using the Bicarbonate dithionite extraction sequentially. Water and sodium hydroxide were included to estimate the full spectrum of the OM quality and concentration. There are some changes in individual cores but not shown as the data was treated as replicate. R1Q3 L100: I did not understand this sentence, and what the reference actually refers to A3. The original study was conducted on water samples and not on any chemical extraction. However, the study cited used it in applying to both water and alkaline extracts hence used as a justification in using it in applying same in our study. R1Q4 L102-104: I did not understand what was done A4. From L102-104, it is established that Fe influences the absorbance at 254 nm on the UV-VIS spectra hence the need to correct that effect. To do that, absorbance at 254 nm of different concentrations of Fe which reflects that of the study was measured and a calibration curve was plotted from that. R1Q5 L109: But does not pH affect fluorescence? A5. We thank the reviewer for this question. We must say from all studies cited so far fluorescence were measured within the scale of pH our study was conducted (pH = 2-13). Presence of metals like Fe quench fluorescence and reducing the pH to 2 eliminates the effect of quenching by metals (Poulin et al., 2014) cited in study. R1Q6 Paragraph at L131: There is a huge difference among treatments, but without further explanation of what they mean, in the Methods, it is hard to appreciate these findings

A6. We appreciate the concern of the reviewer and would improve the clarity in the next stage if it will go further. R1Q7 L154-156: Where is this coming from? No result, or description of the cores was provided to appreciate this statement

A7. Under sampling section (L 68), we stated that the cores were sampled on a longitudinal transects which varied in Fe concentration. Hence this was an explanation to the variation of Fe concentration of the mean. R1Q8 Section 3.2.1: This section is too superficial and is barely interpreted R1Q9 3.2.2: same A8&9. We admit this data wasn't discussed thoroughly this is because there isn't much to say than what have been said. R1Q10 3.3: Here the authors go at length in describing the PARAFAC components, with a level of details that greatly differs from previous sections. I believe the idea is to interpret how specific components may mean a particular chemical composition, which is then used in the following section to interpret associations with Fe, but the ideas are not explicitly connected in my opinion; there is a leap in the level of information provided by the optical approaches in section 3.3 vs the detailed chemical interpretation provided in section 3.4. For example, on what basis, specifically, can you conclude the following: "The structure of both C1 and C2, however, possesses a redox behavior that makes their reaction with Fe-oxyhydroxide reversible leading to the solubilization of the DOM or both Fe and DOM, with the latter subsequently being biodegradable (Burdige, 2007; Keil and Mayer, 2014).". I am not saying that this interpretation is wrong, but I believe that there is not enough information in the manuscript that supports this interpretation based on the authors findings.

A10. The components C1 and C2 have been thoroughly reviewed by (Ishii and Boyer, 2012) and has been cited in the study. This was from an original study by Cory and Mcknight 2005. These were interpreted as reduced and oxidised quinones respectively. The quinone moieties have been established to be taking part in the shuttling of electrons. It was on such basis the data was discussed. R1Q11 L223-225: I do not think these studies have identified C3 specifically (at least they are not presented in Table 1), so this wording is misleading

A11. Yes that is true, they didn't identify C3 as none of those studies used fluorescence, but they identified non-humic components/aliphatic components which is C3 in this study. In fact, in the study by Lalonde et al., 2012, the authors described the component as rich in sugars and proteins. They were cited in the general sense of non-humic quality type of OM. "We also analysed the isotopic composition ($\delta$13C and $\delta$15N) and elemental composition (molar ratio of carbon to nitrogen) of the bulk organic matter and the iron-associated organic carbon fractions of all sediment samples. In most cases, we find that OC-Fe is enriched in 13C ($\delta$13C increases by 1.762.8%; Fig. 2) and nitrogen (C:N decreases by 1.762.8) relative to the rest of the sedimentary organic carbon pool, whereas d15N shows little or no fractionation (Supplementary Figs 1and 2). Natural organic compounds rich in 13C include proteins and carbohydrates25, which are rich in nitrogen and/or oxygen functionalities that favour the formation of inner-sphere complexes with iron." (Lalonde et al., 2012) "We discovered that hematite preferred to sorb more aromatic organic matter as a result of inner-sphere coordination and other interactions, but the aromatic carbon-rich organic matter was more susceptible to the reduction release. These results have important implications for the biogeochemical cycle and stabilization of carbon. First, we provided evidence that iron-bound, non-aromatic carbon was more resistant to reduction reactions, which can preserve aliphatic organic matter." (Adhikari et al., 2015) R1Q12 L228-229: What is "appreciable time", and why is it important? Also, it is weird to end the discussion saying that in the end we are not sure what is going on hence more studies are needed, I would suggest concluding on a more "positive" note in terms of your main contributions to this topic.

A12. We thank the reviewer for his advise and concerns raised. "Appreciable time"

used here is in relation with processes in natural systems. For instance, episodes of anoxia and oxygenation as well as possibility of resuspension do take place. The oscillation of anoxia and oxygenation influences the Fe crystal structure over a period and as well the possible diagenesis of the OM would influence the fate of the OM-Fe. In laboratory experiments devoid of such processes which occurs in the natural environment as in the case of the study cited would probably lead to a different interpretation of the fate of OM-Fe in such study. R1Q13 The conclusion overall is interesting to read, but I found it unsupported by the results and insufficiently developed in key places of the manuscript. I suggest using sentences from the conclusion as key points (maybe paragraphs topics) that guide the development of the introduction and discussion

A13. We are grateful to the reviewer for the recommendations. They will be studied and effected in the revised version of the manuscript. We also are convinced our conclusions are supported by our data discussed so far.

---

## Author Comment (AC2) · 18 Nov 2020

We appreciate the insightful comments offered by the reviewer. Her/his thoughtful review will certainly help us in shaping the manuscript for its intended publication. We are also grateful to the reviewer for noting interesting data in the study. We take the concerns of the reviewer about the data being speculative very seriously and seek to use this opportunity to clarify all those concerns kindly raised by the reviewer. Answers to comments from reviewer One of the confusing sections in the current iteration of the manuscript is that explaining the extraction scheme, the question of redox labile and redox "stable" phases of iron. We admit the lack of clarity in the form that the

extraction scheme was structured in the manuscript. This is to be modified for the sake of clarity. We will seek to clarify it to the best of our ability and hopefully that would clear the doubts the reviewer has enlisted. Firstly, it should be noted that the organic matter (OM) associated with iron (Fe) phases are in particulate and not dissolved form. As the OM-Fe is said to even withstand episodes of anoxia. Also, it is true that all Fe oxyhydroxides would be differentially reduced, however, it is not just the rate of dissolution that differs, but also the degree of dissolution in terms of the percentage dissolved. This points to the fact that some phases are metastable, whiles others are more labile. That was the essential deduction from the result of the extraction scheme implemented. It basically modified the original extraction scheme where the bicarbonate dithionite (BD) is used to dissolve all reactive minerals in a single step reaction over 2 hours. As noted by Jan et al. (2015), few minerals dissolve completely after about 10 minutes, whiles most of the minerals of concern dissolves within 2 hours. Hematite, however, does not dissolve completely even after 6 hours, implying that the usage of a single step extraction scheme overestimates the impact of dissolution of Fe oxyhydroxides. In consequence, a sequential extraction discriminating Fe phases which are more redox labile was considered necessary given that other phases are redox "stable". From such approach, one could infer the impact of dissolution of Fe in the environment. Eusterhues et al., 2014 noticed that in both abiotic and biotic mechanisms, the rate of OM-Fe dissolution and the proportion of the Fe dissolved differs but are comparable. Based on this foundation we characterized the associated OM to understand which quality of OM are associated with the different phases of Fe and discussed it in the light of published and well cited studies. To the issue of the bottom waters: the reservoir used as a natural laboratory is a dimictic system with spring and autumn turnovers. Sampling was done on a longitudinal transect from the inlet to near the dam. Aside the inlet, the other three sampling sites are anoxic over a period. Anoxia is established at the inlet depending on the conditions of the year. Also, the morphology of the base in the reservoir doesn't follow a progressive increase of depth along the longitudinal profile, but an irregular one, hence it is not that prudent to

analyse the data in the longitudinal perspective. With regards to the differences in the DOC and Fe dissolved by both BD1 and 2 there is a clear difference in their means, we admit the standard deviations diminishes the effect of the differences between BD1 and BD2 and this is due to the morphology of the system. Below, we answer specific questions from the reviewer.

Specific comments

R2Q1 (71-3) -These are results and don't belong in this section. Also, what does "more stable" mean?

A1. We have moved this to the results section and "more stable" is a mistake should read "remain constant in the first 2 cm of the vertical profile".

R2Q2 (76) – Why is the bulk scheme mentioned, since as far as I can tell all of the data discussed here comes from the sequential extraction procedure.

A2. It was mentioned to stress why sequential extraction discriminate Fe oxyhydroxides in terms of their redox lability and "stability". Bulk extraction data was placed in the supplementary section.

R2Q3 (78- ) – I may be missing something here, but this sequential extraction scheme does not make sense to me. As I read the text, BD1 extracts Fe that is redox active, then BD2 next extracts iron that is redox stable, and then OH1 next extracts iron that is redox active and finally OH2 extracts iron that is redox stable. If these are sequential extractions and BD1 does not remove all (or most) of the redox active iron why, for example, does any redox active iron escape extraction during BD2 (which is the same as BD1 just longer) to then be extracted by OH1? What am I missing here?

A3. We thank the reviewer for pointing out the confusion. BD1 and BD2 are the only extractants that dissolves reductive Fe. OH1 and OH2 do not dissolve Fe at all especially after the BD extraction. It rather dissolves mainly aluminium (Al) oxyhydroxides. Hence the Fe concentration profiles were from BD extracts. There is no data for Fe

dissolved by NaOH because Fe is not extracted by NaOH extractant. Yes, classically all Fe would be dissolved but it is also a fact that some oxyhydroxides as well as OM-Fe associations are metastable as shown by the data in Eusterhues et al., 2014; Jan et al., 2015.

R2Q4 I'm trying to compare this extraction scheme to others I am a bit more familiar with (e.g., Goldberg, et al. 2012. Chem. Geol. 296, 73-82; Poulton and Canfield 2005. Chem. Geol. 214, 209-221) and am having trouble understanding how NaOH extracts iron oxides. Related to this, it's also not clear to me if any iron is actually extracted in the OH extractions – no data is presented.

A4. NaOH extractant did not dissolve Fe after BD1 and 2. This fraction was not aimed at dissolving OM associated with Fe. The purpose of the NaOH step is to obtain a full spectrum of DOM to determine the quality of OM associated with Fe (see also A6 below).

R2Q5 The concept of redox "active" versus redox "stable" is a bit misleading. All iron oxides will undergo reductive dissolution, but the rates will vary by quite a bit (see, for example, Table 4.1 in Raiswell and Canfield (2012, Geochemical Perspectives 1, 1-220)).

A5. That is true all iron oxides will dissolve, however, the experimental dissolution done in the laboratories with strong reducing agents like BD correspond to longer time scales in the environment. Hematite, even after a period of 6 hours did not dissolve completely. As a result, some Fe oxides are termed metastable and based on that they can be in the environment even in anoxic conditions and would not dissolve completely.

R2Q6 The vast majority of the comparisons made later on in the paper are between the BD1 and BD2 phases (see, for example, the concluding remarks starting on line 247), thus I am further confused by where (and how) the OH extractions fit in here.

A6. As stated in A3 and A5(above), NaOH did not dissolves Fe. The NaOH was used

to have a full spectrum of DOM, for determining the quality of OM associated with Fe. We could also move the NaOH data into supplementary if deemed prudent during the revision.

R2Q7 (137) – Differences in DOC concentration in the BD1 versus BD2 extracts do not necessarily imply differences in OM affinity for these different iron pools. For starters, there is more iron in the BD1 pool so there are presumably more potential iron oxide binding sites for DOM. That may be the simplest explanation for these DOC differences.

A7. Our understanding is that BD1 dissolves Fe oxides which are active compared to that of BD2. Also, Fe oxides dissolved in BD1 had comparatively larger surface areas which favours sorption affinity compared to BD2. We thank the reviewer for the suggesting that there might be an alternative explanation for the differences observed for DOC yields in BD1 v BD2. These will be further reasoned in the next iteration of our manuscript.

R2Q8 (154) – By averaging the results from the 4 cores you are implicitly assuming there is no spatial variability in the sediments along this sampling transect. However, here the authors talk about variations along this longitudinal profile (although no data are shown to support this assertion). Nonetheless, you can't have it both ways. If there is longitudinal variability among the cores you shouldn't be averaging the depth profiles and in fact, by doing so you may be obscuring real depth trends in each core.

A8. We thank the reviewer and appreciate the opinion about the handling of the data. However, we would like to more clearly state, again, that the data was not discussed in a spatial perspective as we are not especially concerned in characterizing the site, but using the sediments from this natural lab for interrogating the OM-Fe association and its fate in general.

R2Q9 (210 – 215) - This is far too speculative and not well supported by the data. Components C1 and C2 may include quinones that can react with iron oxyhydroxides and this may release DOM that may be biodegradable. Yet the profiles presented here

show no evidence of this. Likewise, C3 may be non-redox active and may be irreversibly bound to the iron phases, but I see little to really support this assertion. The points made here in the text are also made in the Abstract (lines 20-21) and the Conclusions (lines 233-6), and are, in my opinion, presented in far too definitive a fashion, given the data presented here. Furthermore, generalization of this speculation to the "rusty iron sink" (line 236) is very premature (at best)

A9. We really appreciate the comments from the reviewer and appreciate where the difficulty may be leading to the comment. The extractant used for the dissolution of Fe oxyhydroxides is a reducing agent. The redox lability and "stability" were assigned to how fast and the proportion of the Iron oxides that will dissolve in the event of a reducing condition as detailed in the general comment section. Now to directly answer the question; As we have stated in the General comment, Fe dissolved in BD1 are usually redox labile those dissolved in BD2. The quality of OM associated in DOM was ascertained to explain how these qualities could be an influential factor in making a phase of the Fe oxides more redox labile than the other. The observed FDOM components have been extensively studied. The shuttling of electrons between Fe oxides and associated aromatic OM have also been studied. It is based on this literature that we discussed our findings, that the identified quality of OM selectively dissolved in the BD1 fraction plays a role the redox lability of Fe phase as established that, such quality of OM enhances the dissolution of the Fe oxyhydroxides. Whiles the protein-like component comparatively doesn't enhance dissolution of Fe explains why they are associated with the Fe phases which are redox "stable".

R2Q10 The differences between the fluorescence characteristics of the DOM associated with the different iron phases is intriguing, but in the context of all of the other issues I have with this manuscript, it's hard for me to be know how to interpret their significance.

A10. We thank the reviewer for giving value to the fluorescence characteristics of the DOM associated with the different iron phases. Clarifications that in our view will help

in interpreting the significance will be included in the revision of our manuscript.